# Stand Characteristics and Soil Properties Affecting the Occurrence of Kunyushan Web-Spinning Sawfly (*Cephalcia kunyushanica* Xiao) in Japanese Red Pine (*Pinus densiflora*) Pure Forests in the Kunyushan Mountains, China

**Ruirui Hu [1], Jun Liang [1,2,*], Xian Xie [1], Yingjun Zhang [2] and Xingyao Zhang [1,2]**

[1] Institute of Forest Ecological Environment and Protection, Chinese Academy of Forestry, Key Laboratory of Forest Protection of State Forestry Administration, Beijing 100091, China; m15712958570@163.com (R.H.); budu39258@163.com (X.X.); xyzhang@caf.ac.cn (X.Z.)

[2] Kunyushan Forest Ecosystem Research Station, Yantai 264100, Shandong, China; kysbhc@163.com

\* Correspondence: liangjun@caf.ac.cn

**Abstract:** The Kunyushan web-spinning sawfly (*Cephalcia kunyushanica*) is a major pest in the Japanese red pine (JRP, *Pinus densiflora*) pure forests in the Kunyushan Mountains of China. In this study, four stand types (ST1–4) were identified in plots of JRP pure forests, based on the pest severity index (PSI; ranging from 0–100). The order of infestation ratio in the four type stands was as follows: ST4 > ST3 > ST2 > ST1. We investigated the correlation of *C. kunyushanica* occurrence with stand characteristics and soil physicochemical properties in the four stand types. The results showed that all stand characteristics were different among the four stand types. Compared with infested plots, healthy (ST1) plots had a higher soil bulk density, and the differences among the groups were significant. Differences in soil water content, non-capillary porosity, and total porosity were significant among the four ST groups. The average organic matter content, total nitrogen (N), and available N were lower in ST1 plots, whereas total potassium (K) was higher compared with other ST groups. In addition, a redundancy analysis suggested that seven (total N, diameter at breast height (DBH), soil water content, bulk density, available K, zinc ion ($Zn^{2+}$), and stem density) of 24 environmental variables were significantly correlated with the ordinations of *C. kunyushanica* occurrence. The results provide theoretical guidance for the ecological control of *C. kunyushanica*, and are also useful for the management of forests in areas where *C. kunyushanica* is a major pest and where site and stand conditions are similar.

**Keywords:** Japanese red pine pure forests; *Cephalcia kunyushanica*; stand type; stand characteristics; soil properties

## 1. Introduction

Rotation times of forests are usually estimated in decades and trees can be infected with a variety of different insect pests and diseases during the juvenile and mature stages of growth [1]. Therefore, suitable forest management practices are needed. Forest managers have traditionally used chemical pesticides rather than non-chemical methods for forest management, which has caused several negative consequences such as the development of pesticide resistance, resurgence of pest populations, emergence of secondary pests, and environmental pollution [2]. Compared with traditional forest management practices, integrated pest management (IPM) is a more effective and environmentally sustainable approach to pest management [3]. In a review of pest management, O'Neil [4] highlighted

that the practice of IPM is often inconsistent with the underlying philosophy. This has resulted in a new concept of ecological control of forest pests [5], which mainly includes biological control, cultivation of insect/pathogen resistant species, site preparation, and the dynamic monitoring of pests. These measures are effective for restricting the growth and development of pests and maintaining balance in the ecosystem [6].

Stand and soil characteristics in a particular site impact the herbivore population at that site. For example, Marchisio et al. [7] reported that severe water stress occurs in the years preceding an outbreak of *Cephalcia arvensis*. The density of pine false webworm (*Acantholyda erythrocephala*) is positively correlated with the relative dominance of host plants and negatively correlated with tree diversity [8]. Site factors (elevation and stand density) and the microclimate of the Japanese red pine (JRP; *Pinus densiflora* Sieb. et Zucc.) forest ecosystem affect the survival, resource selection, and distribution of Kuyushan web-spinning sawfly (*C. kunyushanica*) by affecting the growth potential of host plants or the forest population dynamics [9]. Therefore, studying the effects of stand and soil characteristics on diseases and insect pests is useful for forest managers and is an important component of the ecological control of forest pests.

JRP forests are one of the most economical coniferous forest species in the Kunyushan Mountains of China, where JRP pure forests account for approximately 70% of the vegetation [1,10]. The health and stability of JRP forests are closely related to the development and safety of the local forest ecosystem. Kunyushan web-spinning sawfly (*Cephalcia kunyushanica* Xiao) is a specialized phytophagous insect species [11], which exists only in the Kunyushan Mountains. It was first reported in the Kunyushan Mountains in 1983, and was identified as a new species in 1990 [12]. *C. kunyushanica* can be found on pine stands of all ages, but it prefers the newest needles when the resources are rich. Although it feeds mainly on JRP, it occasionally also feeds on Armand pine (*P. armandi* Franch.), Korean pine (*P. koraiensis* Sieb. et Zucc.), and lodgepole pine (*P. thunbergii* Parl.). *C. kunyushanica* parasitizes pine trees of all ages, with irregular but severe outbreaks in the Kunyushan Mountains. To date, only two studies have been conducted on the biological performance of *C. kunyushanica* [13,14]. Each year, adults of *C. kunyushanica* emerge in May or June and lay eggs in groups on pine needles. Once hatched, larvae spin silk nests; generally, two to four larvae live in one nest [11]. At the base of the nest or in a feces-covered silk tube, the larvae cut and eat needles, and do not disperse actively in the larval stage. The larvae then drill a 5–12 cm deep hole into the tree and prepare to overwinter. Pupation occurs in April of the next year, and adults emerge approximately after one month [13].

In recent years, many studies have been conducted to understand the effect of stand types, site conditions, forest spatial structure, and species connectivity on population of *C. kunyushanica*. However, the influence of soil properties on sawflies has rarely been investigated, the scope of previous research in this area is relatively small, and therefore it does not accurately reflect the influence of JRP stand characteristics on sawfly populations. Further research is needed to advance our understanding of the relationship between stand characteristics and pest severity. This is the first study of the relationship between soil properties and *C. kunyushanica*, and is part of a more comprehensive investigation of pine-insect interactions. The objective of the study was to understand the differences in stand characteristics and soil properties between healthy and infested stands of JRP pure forests. We also inferred the feasibility of ecological approaches to control *C. kunyushanica* densties, based on the effect of environmental factors on the occurrence of *C. kunyushanica*.

## 2. Materials and Methods

### 2.1. Study Area

The Kunyushan Mountains are located in the Jiaodong Peninsula in Shandong Province in eastern China (121°41′34″–121°48′04″ E; 37°11′50″–37°17′22″ N). The climate of this region is moderate due to warm temperate monsoons, with a mean annual temperature of 12.3 °C. The frost-free period ranges from 200–220 days, with a mean annual precipitation of 800–1200 mm and mean annual relative

humidity of 62.6%. The soil is mostly brown sandy loam. JRP trees are the main indigenous conifers in this region, and they are also found in Northeast China, Japan, Korea, and Russia. JRP forests are adapted to a wide range of environments and are naturally distributed from the piedmont to the peak of the mountain (800 m above sea level).

*2.2. Site and Stand Characteristics Survey*

The survey work followed the "Observation Methodology for Long-term Forest Ecosystem Research" of the national standards of the People's Republic of China (GB/T 33027-2016). A total of 121 temporary plots (30 × 30 m) were established systematically in JRP pure forests (from May to August 2017), sharing a similar soil type. The approximate distance between plots was between 30–60 m. The pines were natural secondary forests, covering an area of 15,416 ha in the Kunyushan Mountains. More than 90% of the JRP ages were 32–36 a. The soil had not been treated, such as with tilling or fertilizing. Eight trees were tagged per plot and surrounded by red ropes to serve as temporary markers. Four trees were tagged in the four corners, and another four trees were tagged in the middle of each plot. Trees with a diameter at breast height (DBH) ≥2 cm were identified in each plot, and the height, crown width, and DBH of these trees were recorded. The stem density was expressed as the number of trees per hectare. The basal area at breast height and stand volume in sample plots were calculated according to the following equations [15]:

$$G_{1.3} = \pi D^2/4 \tag{1}$$

$$M = G_{1.3} \times (h + 3)\,fa \tag{2}$$

Here, $G_{1.3}$ is the basal area at breast height (m$^2$·ha$^{-1}$), $D$ represents the DBH (cm), $M$ is the stand volume (m$^3$·ha$^{-1}$), $h$ is the tree height (m), and $fa$ is the experimental form factor (0.42).

*2.3. Determination of the Larval Density of C. kunyushanica*

Web spinning and nesting in branches of JRP trees were used as evidence of infestation by the larvae of *C. kunyushanica*. The infestation ratio (IR) was defined as the proportion of trees in a plot with insect nests (Equation (3)). The status of insect attacks in JRP pure forests in each plot was described by the pest severity index (PSI), which was measured by the five-spot method. Briefly, two trees were selected from each corner and the center of each plot. Insect nests in these ten trees were observed using binoculars or the naked eye. Because each nest usually contained three larvae on average, the larval density per investigated tree was calculated by trebling the nest numbers. The PSI was calculated according to Equation (4), and the pest classification standard is shown in Table 1.

**Table 1.** The pest classification standard of the *C. Kunyushanica* per tree.

| Pest Classification | Value of Corresponding Grade (CGV) | Basis of Classification (larvae·tree$^{-1}$) |
| :---: | :---: | :---: |
| I | 0 | 0 |
| II | 1 | 1–10 |
| III | 2 | 11–30 |
| IV | 3 | 31–50 |
| V | 4 | >50 |

The IR and PSI were calculated according to the following equations:

$$IR\ (\%) = \frac{\sum NTN}{\sum NT} \times 100 \tag{3}$$

$$PSI = \frac{\sum_{i=1}^{10}(NIT \times CGV)}{(\sum NT) \times MV} \times 100 \tag{4}$$

Here, NTN is the number of trees with nests in each plot, NT is the number of trees in each plot, *i* is the number of investigated trees, i.e., 10 per plot, NIT is the number of infested trees, CGV is the value of the corresponding grade, MV is the maximal value of CGV, which was always 4 (Table 1).

## 2.4. Collection and Processing of Soil Samples

To analyze soil physical properties, composite soil samples were collected from each plot at a depth of 0–10 cm using a cutting ring (100 cm$^3$), according to the five-spot sampling method. Soil bulk density (BD, g·cm$^3$) were obtained by using a cutting ring (100 cm$^3$) and calculated as the ratio of the oven-dry soil mass to the cutting ring volume. The soil water content (%) was calculated from the mass loss after oven drying the samples at 105 °C to a constant weight. The natural state of the soils, along with the cutting rings, was weighed ($m_1$, g) after soaking for 12 h in water to estimate the maximum moisture capacity (MMC, %). The cutting rings were then placed on dry sand for 2 h, allowing the non-pore water to be completely drained, then weighted ($m_2$, g). Finally, the soil was sampled from the cutting rings and dried in an aluminum box to a constant weight ($m_0$, g). The equations of maximum moisture capacity, capillary porosity (CP, %), non-capillary porosity (NP, %), and total porosity (TP, %) were as follows [16]:

$$\text{MMC} = \frac{m_1 - m_0}{m_0} \times 100\% \tag{5}$$

$$\text{CP} = \frac{m_2 - m_0}{m_0} \times 100\% \times \text{BD} \tag{6}$$

$$\text{NP} = \left(\text{MMC} - \frac{m_2 - m_0}{m_0} \times 100\%\right) \times \text{BD} \tag{7}$$

$$\text{TP} = \text{CP} + \text{NP} \tag{8}$$

To analyze soil chemical properties, soil samples were collected from a depth of 0–20 cm, also according to the five-spot sampling method, and thoroughly mixed. Impurities in the soil samples, such as stones, animal and plant residues, and other litter, were removed manually. The cleaned soil samples were air-dried and then ground to a final particle size of <2 mm. Soil organic matter (%) was analyzed using the potassium dichromate oxidation-external heating method [17]. The soil pH level was measured in a 1:5 mixture of soil. Total nitrogen (N) was determined using KD310-A distillation and titration unit (OPSIS, Furulund, Sweden) [18]. Total phosphorus (P), total potassium (K), ferric ion ($Fe^{3+}$), cupric ion ($Cu^{2+}$), zinc ion ($Zn^{2+}$), and manganese ion ($Mn^{2+}$) were measured by using microwave digestion system (CEM, Matthews, NC, USA) and plasma emission spectrometer (Thermo, Waltham, MA, USA) [19]. Available nitrogen was monitored by alkaline solution-diffusion method [20]. Available phosphorus was extracted with hydrochloric acid and sulfuric acid solution, while available potassium used ammonium acetate solution for extraction. Then they were monitored with a plasma emission spectrometer (Thermo, Waltham, MA, USA) [21].

## 2.5. Data Analysis

To compare the differences in stand characteristics and soil properties among 121 plots, we divided these plots into four stand types (ST1–4), according to the PSI of each plot: ST1 plots were uninfected (PSI = 0), whereas ST2 (0 < PSI ≤ 20), ST3 (20 < PSI ≤ 40), and ST4 (PSI > 40) plots were infested with *C. Kunyushanica*.

The homogeneity of variance of all indicators were tested before the one-way analysis of variance (ANOVA). The results show that the basal area at breast height, stand volume, bulk density, soil water content, maximum moisture capacity, $Cu^{2+}$, and available K didn't correspond to normal distribution. So, for basal area at breast height, stand volume, and available K, data were transformed by logarithm; for soil water content and maximum moisture capacity, data were dealt with by sine transformation; for $Cu^{2+}$, data was transformed by $1/x$, where $x$ was $Cu^{2+}$ content in each plot; for bulk density, data was transformed by $i^6$, where $i$ was the bulk density in each plot. Then a one-way ANOVA was conducted

to test the difference among four stand types, using Tukey's multiple comparisons. Differences were considered significant at the 5% level of significance. Those were done in SPSS v22.0 (IBM, New York, NY, USA).

The ordination of the occurrence of web-spinning sawflies among the four stand types was determined using CANOCO v5.0 (Microcomputer Power, Ithaca, NY, USA), according to [22]. A detrended correspondence analysis (DCA) showed that the largest gradient length was 1.79 (<3), and therefore redundancy analysis (RDA) was considered an appropriate analytical method [18]. RDA is a multivariate direct gradient analysis method used for a multiple regression analysis of many environmental variables [22]. In this study, the effect of plot averages of 24 environmental variables (including stand characteristics and soil properties) on pest occurrence was investigated in each plot. On the basis of a Monte Carlo permutation test with 499 iterations, the forward selection procedure was used to select environmental variables with $p$-values < 0.05 for the ordination of stand characteristics, and the selected variables were used in final analyses [23,24].

## 3. Results

### 3.1. The Distribution of PSI in Four Stand Types

Table 2 shows the distribution of PSI in four stand types. The number of sawfly-non-infested plots is 34, accounting for 28.10% of the total investigated plots. Among the sawfly-infested plots, the mean values of ST2-ST4 are 11.89, 29.37 and 58.95, respectively. In addition, the table shows the minimum value, maximum value and stand deviation in four stand type.

**Table 2.** The distribution of PSI in four stand types.

|  | Stand Type | Plot Number | Minimum Value | Maximum Value | Mean Value | Standard Deviation |
|---|---|---|---|---|---|---|
| PSI | ST1 | 34 | 0.00 | 0.00 | 0.00 | 0.00 |
|  | ST2 | 32 | 3.13 | 20.00 | 11.89 | 5.55 |
|  | ST3 | 31 | 20.83 | 40.00 | 29.37 | 5.56 |
|  | ST4 | 24 | 43.75 | 91.67 | 58.95 | 13.67 |

PSI is the abbrevation of pest severity index; ST1-4 refer to the four stand types classified according to the pest severity index.

### 3.2. Differences in Stand Characteristics

Table 3 summarizes the stand characteristics in the four stand types of JRP pure forests. The stem density in infested plots was significantly different ($p < 0.01$) form that in healthy plots (ST1), and differences among the four stand types were highly significant. The stem densities in ST3 and ST4 plots were much lower than in ST1 plots. Other growth parameters of trees showed an increasing trend with an increase in PSI values, and the differences in these characteristics among groups were all significant ($p < 0.01$). However, the differences among infested plots (ST2, ST3, and ST4) were non-significant ($p > 0.05$) for tree height, DBH, basal area at breast height, and stand volume. The value of crown width was much higher in ST4 plots than in ST1, ST2, and ST3 plots.

**Table 3.** Stand characteristics of Japanese red pine (JRP) pure forests in different stand types.

| Stand Type | Stem Density (trees·ha$^{-1}$) | Tree Height (m) | Crown Width (m) | DBH (cm) | Basal Area at Breast Height (m$^2$·ha$^{-1}$) | Stand Volume (m$^3$·ha$^{-1}$) |
|---|---|---|---|---|---|---|
| ST1 | 2319 ± 84 a | 5.0 ± 0.3 b | 2.3 ± 0.2 c | 9.8 ± 0.4 b | 8.86 ± 0.76 b | 32.39 ± 4.16 b |
| ST2 | 1991 ± 106 ab | 6.3 ± 0.3 a | 2.9 ± 0.1 bc | 13.6 ± 0.6 a | 16.52 ± 1.31 a | 68.30 ± 6.17 a |
| ST3 | 1910 ± 109 b | 6.5 ± 0.3 a | 3.0 ± 0.1 b | 13.3 ± 0.6 a | 16.90 ± 1.54 a | 69.12 ± 7.52 a |
| ST4 | 1621 ± 88 b | 6.8 ± 0.4 a | 3.7 ± 0.4 a | 13.7 ± 0.7 a | 17.39 ± 1.70 a | 73.47 ± 8.02 a |
| *F*-value | 7.487 ** | 6.605 ** | 11.483 ** | 11.286 ** | 11.145 ** | 8.371 * |

ST1-4 refer to the four stand types classified according to the pest severity index. * $p < 0.05$, ** $p < 0.01$. *F*-value is for comparing across stands. The number after ± is standard error. The different letters indicate significant difference between the stand types, and vice versa.

### 3.3. Differences in Soil Physicochemical Properties

The differences in soil physical and chemical properties among JRP pure forests with different infestation grades are summarized in Tables 3 and 4, respectively. Soil bulk density and soil water content in healthy plots were significantly different ($p < 0.01$) from those in infested plots. Differences in non-capillary porosity and total capillary porosity among the four stand types were significant ($p < 0.01$) and showed an upward trend with the severity of *C. kunyushanica*. No significant differences were detected in non-capillary porosity and total capillary porosity among the infested plots (ST2, ST3, and ST4). No remarkable differences in maximum moisture capacity and capillary porosity were observed among infested plots (Table 4).

**Table 4.** Water-related physical properties of soil in four stand types of JRP pure forests.

| Stand Type | Bulk Density (g·cm$^{-3}$) | Soil Water Content (%) | Maximum Moisture Capacity (%) | Capillary Porosity (%) | Non-Capillary Porosity (%) | Total Porosity (%) |
|---|---|---|---|---|---|---|
| ST1 | 1.22 ± 0.01 a | 10.38 ± 0.41 a | 19.76 ± 0.38 a | 13.35 ± 0.61 a | 6.46 ± 0.32 b | 19.81 ± 0.82 b |
| ST2 | 1.14 ± 0.02 bc | 9.04 ± 0.63 ab | 20.35 ± 0.71 a | 15.22 ± 0.58 a | 7.67 ± 0.48 ab | 22.89 ± 0.72 a |
| ST3 | 1.19 ± 0.02 ab | 8.29 ± 0.46 b | 20.53 ± 0.52 a | 14.87 ± 0.47 a | 8.45 ± 0.32 a | 23.33 ± 0.46 a |
| ST4 | 1.10 ± 0.01 c | 7.44 ± 0.34 b | 19.76 ± 0.83 a | 15.20 ± 0.84 a | 8.08 ± 0.36 a | 23.82 ± 0.99 a |
| *F*-value | 8.124 ** | 4.538 ** | 0.997 | 2.196 | 5.817 ** | 6.730 ** |

ST1-4 refer to the four stand types classified according to the pest severity index. ** $p < 0.01$. *F*-value is for comparing across stands. The number after ± is standard error. The different letters indicate significant difference between the stand types, and vice versa.

An analysis of soil chemical properties revealed strongly significant differences ($p < 0.01$) in the organic matter content, total N, and available K among the four stand types, and significant differences ($p < 0.05$) in available N. There was an increasing trend in the organic matter content, total N, and available N with the level of infestation, with the values of these variables being lowest in ST1 plots and highest in ST4 plots. In contrast, the amount of available K decreased significantly with an increase in PSI values. No significant differences were observed among the four ST groups for total P, total K, available P, various ions ($Cu^{2+}$, $Zn^{2+}$, $Fe^{3+}$ and $Mn^{2+}$), and pH ($p > 0.05$) (Table 5).

**Table 5.** Chemical properties of soil in different stand types of JRP pure forests.

| Stand Type | Organic Matter Content (g·kg$^{-1}$) | Total N (g·kg$^{-1}$) | Total P (g·kg$^{-1}$) | Total K (g·kg$^{-1}$) | Available N (mg·kg$^{-1}$) | Available P (mg·kg$^{-1}$) | Available K (mg·kg$^{-1}$) | Cupric Ion (mg·kg$^{-1}$) | Zinc Ion (g·kg$^{-1}$) | Ferric Ion (g·kg$^{-1}$) | Manganese Ion (g·kg$^{-1}$) | pH |
|---|---|---|---|---|---|---|---|---|---|---|---|---|
| ST1 | 32.37 ± 3.14 b | 0.96 ± 0.07 b | 0.17 ± 0.02 a | 22.84 ± 0.71 a | 74.57 ± 6.06 b | 0.96 ± 0.10 a | 73.89 ± 3.68 a | 3.45 ± 0.29 a | 0.081 ± 0.003 a | 21.61 ± 1.04 a | 0.430 ± 0.023 a | 4.47 ± 0.04 a |
| ST2 | 49.26 ± 3.94 a | 1.51 ± 0.12 a | 0.22 ± 0.03 a | 22.89 ± 0.85 a | 101.71 ± 10.38 ab | 1.20 ± 0.13 a | 67.24 ± 4.06 a | 5.42 ± 1.06 a | 0.087 ± 0.003 a | 24.51 ± 0.97 a | 0.436 ± 0.024 a | 4.39 ± 0.06 a |
| ST3 | 45.52 ± 3.87 ab | 1.66 ± 0.010 a | 0.25 ± 0.02 a | 22.40 ± 0.74 a | 101.84 ± 7.98 ab | 1.20 ± 0.11 a | 66.52 ± 2.56 ab | 6.27 ± 0.95 a | 0.081 ± 0.003 a | 24.52 ± 0.87 a | 0.434 ± 0.020 a | 4.35 ± 0.04 a |
| ST4 | 59.14 ± 4.96 a | 1.72 ± 0.114 a | 0.23 ± 0.02 a | 21.30 ± 0.76 a | 107.18 ± 9.18 a | 1.32 ± 0.20 a | 54.48 ± 2.81 b | 5.78 ± 0.89 a | 0.086 ± 0.002 a | 24.62 ± 0.77 a | 0.465 ± 0.025 a | 4.30 ± 0.04 a |
| *F*-value | 7.701 ** | 11.435 ** | 1.996 | 0.812 | 3.322 * | 1.260 | 5.074 ** | 0.717 | 0.930 | 2.380 | 0.420 | 1.832 |

ST1-4 refer to the four stand types classified according to the pest severity index. * $p < 0.05$, ** $p < 0.01$. *F*-value is for comparing across stands. The number after ± is standard error. The different letters indicate significant difference between the stand types, and vice versa.

### 3.4. Ordination of the Occurrence of C. kunyushanica

The infestation ratio (IR) of *C. kunyushanica* raged from 0–90%. The mean values of IR in ST1-ST4 were 0%, 37.68%, 42.93%, and 67.33%, respectively. IR and PSI were taken as the species variables in the redundancy analysis. Among the 24 environmental variables, seven (total N, DBH, soil water content, bulk density, available K, $Zn^{2+}$ concentration, and stand density) were significantly related ($p < 0.05$) to the occurrence of *C. kunyushanica*. Their effect on the occurrence of *C. kunyushanica* followed the order of total N > DBH > soil water content > bulk density > available K > $Zn^{2+}$ concentration > and stand density (Table 6, Figure 1). The first two components of the RDA axes explained 54.75% of the variance in the relationship between the occurrence of *C. kunyushanica* and the seven selected environmental factors. In addition, the *F*-ratio was 38.45 and the *p*-value was 0.002 (Figure 1), indicating that the linear correlation between the sorting axis and environmental factors reflected the relationship of *C. kunyushanica* with stand and soil factors, and the result of the sequencing was reliable.

**Table 6.** Forward selection of seven significant environmental variables by Monte Carlo permutation test in RDA.

| Variables | Contribution % | *F*-Ratio | *p*-Value (5%) |
|---|---|---|---|
| Total N | 39.9 | 38.4 | 0.002 |
| DBH | 25.5 | 30.7 | 0.002 |
| Soil water content | 6.4 | 8.2 | 0.004 |
| Bulk density | 7.9 | 10.9 | 0.002 |
| Available K | 4.1 | 5.9 | 0.02 |
| $Zn^{2+}$ | 3.5 | 5.3 | 0.02 |
| Stem density | 2.1 | 3.2 | 0.046 |

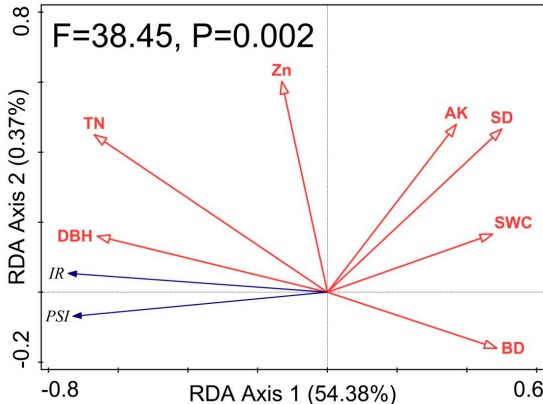

**Figure 1.** First two canonical axes of redundancy analysis (RDA) ordination biplot between the occurrence of web-spinning sawflies and significant environmental variables. SD, stem density; TN, total nitrogen; AK, available potassium; SWC, soil water content; DBH, diameter at breast height; BD, bulk density; Zn, Zinc ion; IR, infestation ratio; PSI, pest severity index.

Forward selection of the seven significant environmental factors in the RDA ordinations showed that the occurrence of *C. kunyushanica* was primarily influenced by total N, DBH, soil water content, bulk density, available K, $Zn^{2+}$, and stem density, and their total contribution to *C. kunyushanica* occurrence was 89.4% (Table 6). The RDA ordination biplot revealed specific associations between the occurrence of *C. kunyushanica* and environmental factors; DBH, total N, and $Zn^{2+}$ were positively associated with IR and PSI, whereas available K, stem density, soil water content, and soil bulk density had a negative association with IR and PSI (Figure 1).

## 4. Discussion

To determine the relationship between stand characteristics, soil properties, and the occurrence of *C. kunyushanica* in the Kunyushan Mountains of China, we examined the differences of these variables in healthy (sawfly-non-infested) and infested stands of JRP pure forests. Stand characteristics of non-sawfly-infested stands of JRP pure forests were different from those of sawfly-infested JRP stands. Stem density was lower, while tree height, crown width, DBH, basal area at breast height, and stand volume were higher in stands infected by *C. kunyushanica*. Soil bulk densities and soil water content were higher in non-sawfly-infested stands than those in sawfly-infested stands. Among the soil chemical properties, the mean values of organic matter, total N, and available N were lowest in healthy plots, while available K was lowest in infected plots.

Although several studies have analyzed the relationship between pest density and stand characteristics, the results have always been ambiguous. In a previous study, Japanese larch (*Larix leptolepis* Gordon) density, tree height, DBH, and the proportion of larch stems were not strongly correlated with the prepupal densities of larch web-spinning sawflies (*C. koebelei* Rohwer) when larch stands had closed canopies [25]. McMillin et al. [26] reported inconsistent effects of stem density on three species of sawflies. In this study, we observed that the PSI increased as the stem density decreased, and *C. kunyushanica* prefers to live in better growing (with higher tree height, DBH, crown width, and so on) JRP forests. Differences in stem densities may have been derived from differences in pest species; for example, the density of *Neodiprion autumnalis* Smith was highest in areas with low tree density and stem density had no influence on *N. xiangyunicus* Xiao and Huang [26]. However, Sun et al. [1] suggested that "*C. kunyushanica* prefers to occur in dense Japanese red pine stands", which is inconsistent with our results. The main reason for this discrepancy may be that Sun et al. used only seven plots, which was not sufficient for the accurate analysis of *C. kunyushanica* in the Kunyushan Mountains. Differences in the size, thickness, and vigor of trees are reflected in the DBH, height, and canopy of trees [27], and these stand characteristics are positively correlated with each other [28]. Therefore, the densities of sawflies were higher in trees with a greater height, canopy width, and DBH than in trees with shorter height, smaller crown width, and DBH. This phenomenon is explained by the plant vigor hypothesis [29]. We showed that *C. kunyushanica* was most likely to occur in large tree stands, which is consistent with the results of De Somviele et al. [30], who reported that variations in outbreak intensity, as measured by defoliation intensity, are positively correlated with mean stand volume, mean stand height, and basal area, but are negatively correlated with stand density. This may be because host plants contain reserves to restore their foliage, food does not become scarce, and thus, do not drive the sawflies out to less preferable plots.

Among the water-related physical properties of soil, our results showed that soil water content in ST4 plots was significantly lower than that in ST1 plots. It is possible that the highest PSI value of ST4 plots was caused by water stress [7]. The association between herbivore outbreaks and sites under water stress may be caused by stress-induced physiological changes in the host, leading to more attractive or acceptable foliage with lower levels of defense compounds. Additionally, these changes enable insects to escape regulation by their natural enemies [31]. Soil bulk density is a key soil parameter that is directly related to many soil properties and processes, including porosity, soil moisture, and erodibility [32]. Soil bulk density can be estimated from the organic matter content of soil, because these variables are negatively correlated [33]. Many abiotic factors affect mortality of pupae in the soil, for examples, teneral *Dacus oleae* suffered a higher mortality in hard soil [34]. The mature larvae of *C. kunyushanica* also overwinter in soil from August to April of the next year [14]. Soil porosity strongly affects the movement of air and water through the soil layers, as well as the soil hardness [35–37], which affects how the mature larvae or pupae survive. The overwintering larvae populations of the previous year are the basis of the adults populations this year. Thus, ST4 plots had the highest PSI, with higher non-capillary porosity and total porosity than other plots.

The ST4 plots had the highest total N and available N, indicating a positive correlation between PSI and these soil variables. It is possible that N enrichment indirectly affects herbivores by changing

the internal microclimate, structure, and quality of host plants, thus changing the behavior and physiological and chemical defense mechanisms of herbivores [38,39]. Increases in the available N content of soil increase the leaf N content, photosynthetic rate, yield, and seed protein content, which enrich the nutrient resources and consequently increase the herbivores populations. This phenomenon is explained by the resource concentration hypothesis [40–43]. Changes in N content also affect host plant allelopathy, which further affects the oviposition and feeding sites of herbivores, and alters their ability to escape predators [44,45]. A positive correlation has been reported between leaf N concentration and insect survival, development, growth, and reproduction [46]. Cheng et al. [47] reported that the total N content is positively correlated with the organic matter content, which improves the availability of soil nutrients. So, similarly to the nitrogen nutrients, soil organic matter enriches the nutrient resources and consequently increases the herbivores populations. Besides, Nemer et al. [48] showed that the mortality of *C. tannourinensis* prepupae is 100% in sandy soils, which lack organic matter. This also suggests that soil organic matter restricts the development and survival of the pupae of Pamphiliidae insects. The PSI of *C. kunyushanica* increased as the level of available K decreased, which can be explained by the plant stress hypothesis [49]. This suggests that the number of herbivores increases with increased translocation of nutrients in host plants due to environmental stress [50]. In our study, it was possible that the deficient potassium nutrition of the JRP could have enhanced nitrogen nutrition available to *C. kunyushanica*.

The RDA ordination graph indicated that different variables either positively or negatively influenced stand characteristics. Seven significant environmental factors affected the occurrence of *C. kunyushanica* by affecting the growth of JRP trees. The RDA was useful because it not only simplified the number of variables effectively but also determined the independent contribution rate of each variable to the environment. Moreover, RDA describes the explanatory ability of specific indicators and enables reliable quantitative ranking [22,51]. In this study, we used RDA to explore the relationship between environmental factors and *C. kunyushanica* and intuitively explained the interaction among multiple variables.

The ecological control of forest pests can make use of various ecological factors, including pest themselves, to control the structure and function of ecosystem, and reach the goal of pest control-sustainable forestry development [6]. From a practical standpoint, our results strongly suggest that control of *C. kunyushanica* through ecological approaches can be achieved by increasing JRP density appropriately. Meanwhile, the overwintering larvae or pupae populations of the previous year are the basis of the adults populations this year. Thus, reducing the number of larvae or pupae may be a potential approach for preventing sawfly outbreaks. Our study shows *C. kunyushanica* usually occurs in soils with higher levels of organic matter and nitrogen. Thus, in these places, we can destroy the soil chamber constructed by the mature larvae so as to prevent them from completing pupation.

## 5. Conclusions

This study revealed differences in the stand characteristics and soil properties among healthy JRP pure forests and those infested with *C. kunyushanica*. The results may be useful for the management of forests in areas where *C. kunyushanica* is a major pest and where site and stand conditions are similar. Among the 24 factors, stem densities, DBH, total N, $Zn^{2+}$, available K, soil water content, and bulk density are the most significant factors affecting *C. kunyushanica* density. Therefore, the seven factors should be considered for controlling the *C. kunyushanica* population in the future, such as adjusting the stem density. In addition, this study was conducted at the stand scale. We propose that the further research is needed to explore the relationship between stand factors and the occurrence of *C. kunyushanica* at tree scale, thus using roadside sampling methods to cover large areas in a cost-effective approach.

**Author Contributions:** R.H. and J.L. conceived and designed the experiments; R.H. and X.X. performed the experiments; Y.Z. and X.Z. conducted the data analysis.

**Funding:** This research was supported by the National Key Research and Development Project of China (2017YFD0600104), the CFERN & BEIJING TECHNO SOLUTIONS Award Funds on excellent academic achievements and the National Natural Science Foundation of China (31270682).

**Acknowledgments:** We thank Jun Wang, Xiaowen Yuan, and Xin Song for their help in collecting and handling the vast amount of data.

**Conflicts of Interest:** The authors declare no conflict of interest.

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
