# Peer review of "Stand Characteristics and Soil Properties Affecting the Occurrence of Kunyushan Web-Spinning Sawfly (Cephalcia kunyushanica Xiao) in Japanese Red Pine (Pinus densiflora) Pure Forests in the Kunyushan Mountains, China"

_forests, doi:10.3390/f9120760_

Round 1

Reviewer 1 Report

Very interesting. I suggest you add more specifics to your table and figure headers to make them stand alone. I think you could remove Table 6 and add another paragraph in the conclusions about how the results apply to management.

Forests Review Article Edits

Stand Characteristics and Soil Properties Affecting the Occurrence of Web-spinning Sawfly (Cephalcia kunyushanica) in Japanese Red Pine (Pinus densiflora) Pure Forests in the Kunyushan Mountains, China

Line 33 remove the word “be”

Abstract: IR and RDA are only mentioned one time, so you do not need to abbreviate them in the abstract, add the abbreviation to the first time you mention them in the body of the article.

Line 54 please put the Latin name in parenthesis after pine false webworm

Line 57 is where you first mention web spinning sawfly, so please put the scientific name in parenthesis after it there.

Line 64, 68 you can put the genus as C. instead of Cephalcia

I would not make line 83 a new paragraph, but suggest combine paragraph starting at line 77 with line 83.

Line 86-87 The sentence “We investigated the effect of stand …”, is very similar to the previous sentence and is not needed. Also, the pest species is now referred to as sawfly, if you are to do this throughout the paper then introduce the species in line 57 as such. I.E. web spinning sawfly (Cephalcia kunyushanica, sawfly)

Line 97 in noting that the pine is indigenous to this Chinese region you may also want to explain that the trees range is also a part of Japan, Korea, NE China.

Line 139, was the soil chemical properties samples collected with the five-spot sampling method?

Line 152, did you have to transform your data for normality in the ANOVA? If so, mention what transformation.

Line 154, should cite Excel and SPSS makers as you did in line 157 for CANOCO

Line 167, this is a long sub-heading, you could delete “among different stand types of JPR pure forests”

Line 168, would sound better if you removed the word “changes”

Line 177, Line 197, 199, 203, 209 suggest you add more specifics to your table and figure headers, such as restating what ST1-4 means, that the F-value is for comparing across stands.

Line 180, this is a long sub-heading, you could delete “among different stand types of JPR pure forests”

Table 4. needs some formatting done to it so that it lines up like the other tables (if possible, can the font be decreased?)

Line 204, IR had already been defined, so you can delete “infestation ratio”

Line 211, Is this to be a footer? Could just put it directly into Table 5. Description

Line 215, I’m not sure Table 6 is needed. I have not seen this in other papers, although it could be added in an appendix.

Line 235-236, I would recommend shortening the sentence by ending it after “89.4% (Table 5)

Line 241-242, I suggest changing this to a few sentences, adding the key results you plan to discuss

Lines 243-245 I suggest removing this sentence

Line 277, do you mean “enabling the development of healthy larvae and pupae”?

Line 329-330, the last sentence has me expecting another paragraph on how the results can guide forest management.  It would be good to change it if you are not going to include management recommendations, or it would be just as good to add another paragraph about management.

Author Response

Response to Reviewer 1 Comments

Thank you for your review and working on our manuscript: forests-390329 (Stand Characteristics and Soil Properties Affecting the Occurrence of Web-spinning Sawfly (Cephalcia kunyushanica) in Japanese Red Pine (Pinus densiflora) Pure Forests in the Kunyushan Mountains, China). We have made some modification point-by-point according to your comments. We deeply appreciate your consideration of our manuscript. If the article still needs some further modification, please let us known. We will do our best to revise it.

Very interesting. I suggest you add more specifics to your table and figure headers to make them stand alone. I think you could remove Table 6 and add another paragraph in the conclusions about how the results apply to management.

Thank you for your interest in our research. We have supplemented the information, such as restating what ST1-4 means, that the F-value is for comparing across stands. But it was in the form of footnotes below each table. Because the contents of Table 6 and Fig. 1 (original Fig. 2) are somewhat repetitive. We have removed the Table 6 and put the explanatory rates of Axis1 and Axis2 as well as the F and P values in figure1 (original Fig. 2). We have discussed the significant of our results by ecology control of forest pest in “Discussion” part. In addition, we revised the conclusion.

Forests Review Article Edits

Point 1: Line 33 remove the word “be”

Response 1: We have removed the word “be” in L33.

Point 2: Abstract: IR and RDA are only mentioned one time, so you do not need to abbreviate them in the abstract, add the abbreviation to the first time you mention them in the body of the article.

Response 2: Thank you for your suggestion. We have removed “IR” and “RDA” in abstract and added them to the first time mentioned in the text.

Point 3: Line 54 please put the Latin name in parenthesis after pine false webworm

Response 3: I’m sorry to make such a mistake, and we have add Acantholyda erythrocephala after pine false webworm.

Point 4: Line 57 is where you first mention web spinning sawfly, so please put the scientific name in parenthesis after it there.

Response 4: We have put C. kunyushanica after kuyushan web spinning sawfly.

Point 5: Line 64, 68 you can put the genus as C. instead of Cephalcia

Response 5: We have put the genus as C. instead of Cephalcia.

Point 6: I would not make line 83 a new paragraph, but suggest combine paragraph starting at line 77 with line 83.

Response 6: Thank you for your suggestion. We have combined paragraph starting at line 77 with line 83, which made the whole paragraph more coherent and complete.

Point 7: Line 86-87 The sentence “We investigated the effect of stand …”, is very similar to the previous sentence and is not needed. Also, the pest species is now referred to as sawfly, if you are to do this throughout the paper then introduce the species in line 57 as such. I.E. web spinning sawfly (Cephalcia kunyushanica, sawfly)

Line 97 in noting that the pine is indigenous to this Chinese region you may also want to explain that the trees range is also a part of Japan, Korea, NE China.

Response 7: We have removed the sentence “We investigated the effect of stand …” in L 86-87. In line 57, and we put C. kunyushanica after kuyushan web spinning sawfly, then we used C. kunyushanica to express kuyushan web spinning sawfly in the full text. You are really knowledgeable. Japanese red pine is naturally distributed in eastern Heilongjiang, Changbaishang Mountain in Jilin, central Liaoning to Liaodong Peninsula of Liaoning Province, and Jiaodong Peninsula of eastern Shandong in China, and it is also found in Japan, Korea and Russia. So, we have added “and it is also found in Northeast China, Japan, Korea and Russia” after “JRP trees are the main indigenous conifers in this region”.

Point 8: Line 139, was the soil chemical properties samples collected with the five-spot sampling method?

Response 8: Yes, the soil chemical properties samples collected with the five-spot sampling method, and we have added it in L139.

Point 9: Line 152, did you have to transform your data for normality in the ANOVA? If so, mention what transformation.

Response 9: I’m sorry we didn’t do the homogeneity of variance test. However, in the revised manuscript, we have examined assumptions of ANOVA. The homogeneity of variance of all indicators were tested before the one-way analysis of variance (ANOVA). The results show that the basal area at breast height, stand volume, bulk density, soil water content, maximum moisture capacity, Cu2+, and available K didn’t accord with normal distribution. So, for basal area at breast height, stand volume, and available K, data were transformed by logarithm; for soil water content and maximum moisture capacity, data were dealt with sine transformation; for Cu2+, data was transformed by 1/x, where x was Cu2+ content in each plot; for bulk density, data was transformed by i6, where i was the bulk density in each plot.

Point10: Line 154, should cite Excel and SPSS makers as you did in line 157 for CANOCO

Line 167, this is a long sub-heading, you could delete “among different stand types of JPR pure forests”

Line 168, would sound better if you removed the word “changes”

Response 10: We have cite the maker of SPSS, but because we only used Excel for data calculation and run ANOVA by SPSS, we deleted Excel in L157.  We have delete “among different stand types of JPR pure forests” in L167 and removed the word “changes” in L168.

Point11: Line 177, Line 197, 199, 203, 209 suggest you add more specifics to your table and figure headers, such as restating what ST1-4 means, that the F-value is for comparing across stands.

Response 11: We have supplemented the information, such as restating what ST1-4 means, that the F-value is for comparing across stands. But it was in the form of footnotes below each table.

Point12: Line 180, this is a long sub-heading, you could delete “among different stand types of JPR pure forests”

Response 12: We have delete “among different stand types of JPR pure forests” as you suggested.

Point13: Table 4. needs some formatting done to it so that it lines up like the other tables (if possible, can the font be decreased?)

Response 13: We tried to change Table 5 (original Table 4) into the same format as the other tables by decreasing the font, but due to too many factors, the font would not fit. Then the font would be unclear if we keep decreasing. So we're going to keep the original format, and it's a little bit easier to see. Thank you for your comments as well, and we will pay attention to them in future writing.

Point14: Line 204, IR had already been defined, so you can delete “infestation ratio”

Response 14: We have delete “infestation ratio” in line 204.

Point15: Line 211, Is this to be a footer? Could just put it directly into Table 5. Description

Response 15: Yes, the sentence in L211 is a footer of Table 5, and I have put 5% into Table 5 as you advised.

Point16: Line 215, I’m not sure Table 6 is needed. I have not seen this in other papers, although it could be added in an appendix.

Response 16: Your suggestion is very good, because the contents of Table 6 and Fig. 1 (original Fig. 2) are somewhat repetitive. We have removed the Table 6 and put the explanatory rates of Axis1 and Axis2 as well as the F and P values in figure1 (original Fig. 2).

Point17: Line 235-236, I would recommend shortening the sentence by ending it after “89.4% (Table 5)

Response 17: We have deleted the sentence after “89.4%” as you suggested.

Point18: Line 241-242, I suggest changing this to a few sentences, adding the key results you plan to discuss

Response 18: Another reviewer raised similar suggestions about the discussion, and I revised them in combination with your comments. First, the research objective are given. Then, the key results planned to discuss are added.

Point19: Lines 243-245 I suggest removing this sentence

Response 19: We have removed that sentence in L243-245 as you suggested.

Point20: Line 277, do you mean “enabling the development of healthy larvae and pupae”?

Response 20: I’m sorry to make you obscure. We have revised it asenabling the development of larvae and pupae".

Point21: Line 329-330, the last sentence has me expecting another paragraph on how the results can guide forest management. It would be good to change it if you are not going to include management recommendations, or it would be just as good to add another paragraph about management.

Response21: Thank you for your suggestion about supplementing information on forest management. We have discussed the significant of our results by ecology control of forest pest in “Discussion” part. In addition, we revised the conclusion.

Reviewer 2 Report

Hu et al. have studied effects of  stand and soil characteristics on occurrence of Cephalcia kunyushanica in Pinus densiflora forest area. The topic is interesting and novel, because there is limited data of C. kunyushanica performance in general, and knowledge of factors affecting its occurrence is scarce. The study is interesting for readers interested in forest herbivores in general and their ecological control. The topic is also important from point of climate change, because insect outbreaks are predicted to increase due to warming. 

The main strength of the study is using the large field data which makes the study ecologically relevant. The study contains several inaccuracies and some of the statistical analyses need to be done again. I also think that bias in sampling and results affected your conclusions. I will point the minor and major issues when I go through the manuscript in detail.

Abstract. L27-28. You should not describe the difference in PSI between stand types in the middle of the results, but earlier around the L18-19.

Remove P-values from the abstract. I even suggest removing expressions 'Statistically significant', because readers would assume you only report significant results.

Introduction. 

The introduction gives an excellent background for the study and rise high hopes for the discussion and conclusions, which will be commentend later.

L76. Is there information on how needle age affects feeding or oviposition? Some other sawflies are known to prefer newest needle generation, some other feed all generations etc.

Go carefully through your whole manuscript for the scientific and common names of insect and plant species. When new a species is mentioned for the first time, the scientific name should be mentioned. Be consistent in using common or scientific names. A few examples below:

L54. Scientific name for one species and common name for the other species at first mention.

L57.  Your own study species mentioned for the first time using a common name.

L248. scientific name missing for larch

Material and methods

Section 2.2 (L100)

How large was the area studied in square kilometres? What was the average/approximate distance between the plots? How did you choose the plots (systematically, randomly)? This information may be found from the national standards you refer to, but you need to explain these for the international readers. How many plots were eventually studied? 136 plots was established, but you only processed data from 121 plots (L147.) What was the reason abandoning data from 15 plots?

What was the average age of the trees? Was the tree age uniform in the area? Just to confirm, pines in the area were of natural origin? Was any harvesting or soil preparation (tilling, fertilization) done in the area? 

Section 2.3 (L114)

Table 2 shows that trees were smaller in the stand type I and the tree density was also highest there. This made me think that the observations and results may be biased. As the forest was denser in ST1, did it affect nest calculation? I would assume that it is more difficult to observe all nests from the dense forests. In addition, trees catecorized to ST1 were smaller, and had smaller crown. Can the results just be explained by the amount of foliage: there is more nests in the large crown? I suggest using paramaters describing amount of foliage as covariates in the statistical analyses. 

Table 1. You could add shortening 'CGV' in the second column in the table.

Does 'head' mean number of larvae or number of nests? Specify it in the table.

L130. You could write MV of the maximal value of CGV

Section 2.4 L132

You must describe methods even shortly. Your national standards and laboratory practises are unknown for international readers. For example, it would be important to know if you measured soil dry or wet bulk density.

Section 2.5 L146

L149-151. You divided plots to four classes according to infestation. A highly important information is missing: how many plots belonged to each class? 

Give PSI average and standard deviation for the whole study area, as well as maximum value.

L152-155. Did you examine assumptions of ANOVA? Explain how. Figure 1 suggests that assumptions of ANOVA for homogeneity of variances was not fullfilled. Make pairwise comparisons again using appropriate post hoc test (e.g. Tukey). LSD should not be used, because it does not make correction for multiple comparisons, and thus, gives significant p-values too easily.

L154. No need to mention Excel, if you only used it for data calculation and sorting and run ANOVA by SPSS.

L161-L163. I am not native English speaker, but the information of the sentence sounds wrong. I think you mean that you used plot averages of 24 variables to investigate their effect on pest occurrence.

Results

Table 2. Add information of number of plots per ST. If possible, add also average age of the trees.

Tables 2-4. What is the number after +? P-values are never exact. Correct as P < 0.05, P < 0.01. Add the information below the Table also below the Tables 3-4. Remove the text 'The same below' from Table 2. 

L170 and throughout the Results. There is no need to repeat F and P -values in the text, if they are found in the Tables or figures.

L171. and L188 and L325. Remove interpretative and descriptive words and use statistical terms.

L172. Replace the expression 'growth vitality' for example by 'other growth parameters'. Expression is interpretative and discussive. Moreover, the results may not represent vitality. (I will come to that later.)

L188-188. Put the result of bulk density (ST4 vs ST1) after you mention that parameter for the first time on L183.

Fig. 1. This figure is  not a good scientific style and is directly copied from SPSS output. What does the whiskers, upper and lower edge of the boxes and horizontal line in the box indicate?  Please explain these or preferably replace with a figure showing mean and the same variation  as in Tables 2-4.

Fig. 2. I assume you have an error in the figure. PI should be IR?

Discussion

L241-245. The discussion does not start fluently. First you give a result and then you tell your aim. Please modify.

The trees catecorized to ST1-ST4 differed in growth so that healthy plots were the smallest and densest. I was earlier asking if these differences could cause observation bias and explain the results. I am also asking  if the trees at ST1 were just younger and growth is therefore smaller. Since we don't know the age of the trees, I strongly disagree using expressions vigor factors (L252) or healthy (L254). I also think that better nutrient availability (mainly N) partly explain better growth of the sites ST2-ST4. It does not mean that forest stands ST1 is not healthy even if it does not grow as fast as the other. 

L250. I did not understand the sentence, especially 'after canopy closure'

L276-279 and L293-294 you make big leaps from soils to insects. Soils affect first trees and tree quality affects insect performane. Thus, mention the role of trees, too.

L299 meaning of 'form of phloem N available' is not undestandable, please explain

L299-301. Remove. No need to repeat the results and speculate with non-significant results. 

L302-314. Shorten this chapter. You should not repeat the results here. You also discuss about the same points as earlier in the discussion, and you should combine them. You can shortly mention the benefit of RDA, but it is not necessary.

Conclusions. I think this is not a conclusion, but summary of results. Please, re-write.

After your excellent introduction, I was waiting for discussion about the significance of the results from point of forest management and IPM from the modern perspective. I encourage you to discuss of these aspects, even shortly.

Author Response

Response to Reviewer 2 Comments

Hu et al. have studied effects of stand and soil characteristics on occurrence of Cephalcia kunyushanica in Pinus densiflora forest area. The topic is interesting and novel, because there is limited data of C. kunyushanica performance in general, and knowledge of factors affecting its occurrence is scarce. The study is interesting for readers interested in forest herbivores in general and their ecological control. The topic is also important from point of climate change, because insect outbreaks are predicted to increase due to warming.

The main strength of the study is using the large field data which makes the study ecologically relevant. The study contains several inaccuracies and some of the statistical analyses need to be done again. I also think that bias in sampling and results affected your conclusions. I will point the minor and major issues when I go through the manuscript in detail.

Thank you for your interest in our research and your review on our manuscript: forests-390329 (Stand Characteristics and Soil Properties Affecting the Occurrence of Web-spinning Sawfly (Cephalcia kunyushanica) in Japanese Red Pine (Pinus densiflora) Pure Forests in the Kunyushan Mountains, China). We have made some modification point-by-point according to your comments. We deeply appreciate your consideration of our manuscript. If the article still needs some further modification, please let us known. We will do our best to revise it.

       As for the statistical analyses, we have done the homogeneity of variance of all indicators before the one-way analysis of variance (ANOVA), and revised some results in Tables correspondingly. As for “bias in sampling and results affected your conclusions”, I will answer it detailedly in Response 7.

Point 1: Abstract. L27-28. You should not describe the difference in PSI between stand types in the middle of the results, but earlier around the L18-19.

Response 1: We have adjusted the sentence “and the order of infestation ratio in the four type stands was as follows: ST4 > ST3 > ST2 > ST1” after “based on the pest severity index (PSI; ranging from 0–100)” as you advised.

Point 2: Remove P-values from the abstract. I even suggest removing expressions 'Statistically significant', because readers would assume you only report significant results.

Response 2: Thank you for your suggestion so that let us know what we should pay attention to in the abstract. We have removed P-values and removed expressions 'Statistically significant' as much as possible.

Introduction.

Point 3: The introduction gives an excellent background for the study and rise high hopes for the discussion and conclusions, which will be commentend later.

L76. Is there information on how needle age affects feeding or oviposition? Some other sawflies are known to prefer newest needle generation, some other feed all generations etc.

Response 3: I have added the information about the effect of needle age on feeding of sawflies after “negatively correlated with tree diversity [8]”. The larvae of Diprion pini, a common pine sawfly in Finnish pine forests, feed on all needle age classes of the host trees. C. kunyushanica are also can be found on pine stands of all ages but they prefer newest needle when the resource is rich.

Point 4: Go carefully through your whole manuscript for the scientific and common names of insect and plant species. When new a species is mentioned for the first time, the scientific name should be mentioned. Be consistent in using common or scientific names. A few examples below:

L54. Scientific name for one species and common name for the other species at first mention.

L57. Your own study species mentioned for the first time using a common name.

L248. scientific name missing for larch

Response 4: I'm sorry for making those mistakes, and we have revised them in corresponding position.

Material and methods

Point 5: Section 2.2 (L100)

How large was the area studied in square kilometres? What was the average/approximate distance between the plots? How did you choose the plots (systematically, randomly)? This information may be found from the national standards you refer to, but you need to explain these for the international readers. How many plots were eventually studied? 136 plots was established, but you only processed data from 121 plots (L147.) What was the reason abandoning data from 15 plots? 

Response 5: Thank you for your suggestion so that we can supplement more information on site and stand characteristics. The pines were natural secondary forests, covering an area of 15,416 ha in the Kunyushan Mountains. The approximate distance between plots was between 30 - 60 m. In our research, in order to make stand and site factors in each plot as uniform as possible, we investigated systematically. I’m sorry to make you confused about the number of plots. We abandoned 15 plots because their stand and site factors had high approximation and strong repeatability. That, there were 121 plots eventually studied.

Point 6: What was the average age of the trees? Was the tree age uniform in the area? Just to confirm, pines in the area were of natural origin? Was any harvesting or soil preparation (tilling, fertilization) done in the area?

Response 6: The pines were natural secondary forests. As a result of previous forest managements, more than 90% of the JRP ages were 32 ~ 36 a. The soil had not been treated, such as tilling or fertilizing.

Section 2.3 (L114)

Point 7: Table 2 shows that trees were smaller in the stand type I and the tree density was also highest there. This made me think that the observations and results may be biased. As the forest was denser in ST1, did it affect nest calculation? I would assume that it is more difficult to observe all nests from the dense forests. In addition, trees categorized to ST1 were smaller, and had smaller crown. Can the results just be explained by the amount of foliage: there is more nests in the large crown? I suggest using paramaters describing amount of foliage as covariates in the statistical analyses.

Response 7: Thank you for reminding us of this situation. We also considered the effect of density, so we observed it very carefully during the investigation, and the number of nests in all plots were observed by one person. Japanese red pine is an intolerant tree species, which is greatly affected by the stem density. When the stem density is high, pines will decrease or even avoid overlap to get more light. In addition, the JRP ages were 32 ~ 36 a, belonging to young forests, so we did not find that the overlap between the branches was so great that the results were seriously affected.

There is a phenomenon in which pests prefer to appear in low-density stands. E.G. In reference “Spatial distribution patterns of pine sawflies (Hymenoptera: Diprionidae) in Arizona, US and Sichuan, PR of China”, the result showed that proportionately more egg of Neodiprion autumnalis (a pine sawflies species) were found in the top of crowns at low stand densities. Another reference of “Effect of host tree density and apparency on the probability of attack by the pine processionary moth” showed that the percentage of trees attacked by pine processionary moth (Thaumetopoea pityocampa) was higher in older stands, which had a lower tree density.

Previous studies such as “Stand structure interacts with previous defoliation to influence herbivore fitness” showed that the survival and performance of balsam fir sawfly (Neodiprion abietis) increased with the increased foliar growth. Our study showed more nests in bigger crown width. So, the effect of crown width and foliage on the herbivore population showed the same trend. There is a positive correlation between the amount of foliage and the crown width. So the crown width can be used to replace the amount of foliage to reflect the influence on larvae density.

Point 8: Table 1. You could add shortening 'CGV' in the second column in the table.

Response 8: We have add shortening 'CGV' in the second column in the table as you advised.

Point 9: Does 'head' mean number of larvae or number of nests? Specify it in the table.

Response 9: Here, head means number of larvae , so we have revised the unit as larvae•tree-1.

Point 10: L130. You could write MV of the maximal value of CGV.

Response 10: We have written MV is the maximal value of CGV.

Point 11: Section 2.4 L132

You must describe methods even shortly. Your national standards and laboratory practises are unknown for international readers. For example, it would be important to know if you measured soil dry or wet bulk density.

Response 11: I am sorry that we overlooked this point and did not write down the measurement method. In section 2.4, we have added the methods about measuring soil physical and chemical properties, respectively.

Point 12: Section 2.5 L146

L149-151. You divided plots to four classes according to infestation. A highly important information is missing: how many plots belonged to each class? Give PSI average and standard deviation for the whole study area, as well as maximum value.

Response 12: We have combined the two questions above to revise. The information about PSI was described by Table 2 in text.

Table 2. The distribution of PSI in four stand types.

Stand Type

Plot Number

Minimum Value

Maximum Value

Mean Value

Standard Deviation

PSI

ST1

34

0.00

0.00

0.00

0.00

ST2

32

3.13

20.00

11.89

5.55

ST3

31

20.83

40.00

29.37

5.56

ST4

24

43.75

91.67

58.95

13.67

Point 13: L152-155. Did you examine assumptions of ANOVA? Explain how. Figure 1 suggests that assumptions of ANOVA for homogeneity of variances was not fullfilled. Make pairwise comparisons again using appropriate post hoc test (e.g. Tukey). LSD should not be used, because it does not make correction for multiple comparisons, and thus, gives significant p-values too easily.

Response 13: I’m sorry we didn’t do the homogeneity of variance test. However, in the revised manuscript, we have examined assumptions of ANOVA. After consulting the reference and combining your opinions, we found LSD is sensitive so that even small differences in the means between two levels can be tested. In addition, Tukey test is mainly used for multiple comparisons of 3 or more groups. So, we have replace LSD with Tukey.

The descriptions are as follows. The homogeneity of variance of all indicators were tested before the one-way analysis of variance (ANOVA). The results show that the basal area at breast height, stand volume, bulk density, soil water content, maximum moisture capacity, Cu2+, and available K didn’t accord with normal distribution. So, for basal area at breast height, stand volume, and available K, data were transformed by logarithm; for soil water content and maximum moisture capacity, data were dealt with sine transformation; for Cu2+, data was transformed by 1/x, where x was Cu2+ content in each plot; for bulk density, data was transformed by i6, where i was the bulk density in each plot. Then a one-way ANOVA was conducted to test the difference among four stand types, using Tukey’s multiple comparisons. As for the modification of figure 1, I'll answer it later.

Point 14: L154. No need to mention Excel, if you only used it for data calculation and sorting and run ANOVA by SPSS.

Response 14: We have removed Excel in this sentence.

Point 15: L161-L163. I am not native English speaker, but the information of the sentence sounds wrong. I think you mean that you used plot averages of 24 variables to investigate their effect on pest occurrence.

Response 15: Your understand about the meaning of that sentence is right, and it has been revised.

Results

Point 16: Table 2. Add information of number of plots per ST. If possible, add also average age of the trees.

Response 16: The numbers of plots can be seen in Response 12 (Table 2). The JRP ages investigated were 32 ~ 36 a, belonging to young forests. So we thought of them as being relatively consistent in age, not as independent variables.

Point 17: Tables 2-4. What is the number after+?P-values are never exact. Correct as P < 0.05, P < 0.01. Add the information below the Table also below the Tables 3-4. Remove the text 'The same below' from Table 2.

Response 17: The number after ± is standard error. The information below each table was supplemented or revised as: ST1-4 refer the four stand types classified according to the pest severity index.* P < 0.05, ** P < 0.01. F-value is for comparing across stands. The number after ± is standard error. The same letter indicates that the difference between the means is not significant, and vice versa.

Point 18: L170 and throughout the Results. There is no need to repeat F and P -values in the text, if they are found in the Tables or figures.

Response 18: We have removed the F and P -values in the text.

Point 19: L171. and L188 and L325. Remove interpretative and descriptive words and use statistical terms.

Response 19: Those sentences have been revised in text as you advised.

Point 20: L172. Replace the expression 'growth vitality' for example by 'other growth parameters'. Expression is interpretative and discussive. Moreover, the results may not represent vitality. (I will come to that later.)

Response 20: We have replace the expression 'growth vitality' by 'other growth parameters' in the text.

Point 21: L188-188. Put the result of bulk density (ST4 vs ST1) after you mention that parameter for the first time on L183.

Response 21: Sentence in L188-188 has been removed, because the meaning has been described in L183.

Point 22: Fig. 1. This figure is not a good scientific style and is directly copied from SPSS output. What does the whiskers, upper and lower edge of the boxes and horizontal line in the box indicate? Please explain these or preferably replace with a figure showing mean and the same variation as in Tables 2-4.

Response 22: We thought your point on “figure1 is not a good scientific style” is right, and you let us preferably replace with a figure showing mean and the same variation as in Tables 2-4. Here, we want to express that the data of infestation ratio (IR) was independent and we used it in RDA just as species variables like PSI. So we have deleted Fig.1 and described its information in “3.4 Ordination of the Occurrence of C. kunyushanica.

Point 23: Fig. 2. I assume you have an error in the figure. PI should be IR?

Response 23: I’m sorry to make that mistake. IR is the abbreviation of infestation ratio, and it has been revised.

Discussion

Point 24: L241-245. The discussion does not start fluently. First you give a result and then you tell your aim. Please modify.

Response 24: Another reviewer raised similar suggestions about the first paragraph of discussion, and I revised them in combination with your comments. First, the research objective are given. Then, the key results planned to discuss are added.

Point 25: The trees categorized to ST1-ST4 differed in growth so that healthy plots were the smallest and densest. I was earlier asking if these differences could cause observation bias and explain the results. I am also asking if the trees at ST1 were just younger and growth is therefore smaller. Since we don't know the age of the trees, I strongly disagree using expressions vigor factors (L252) or healthy (L254). I also think that better nutrient availability (mainly N) partly explain better growth of the sites ST2-ST4. It does not mean that forest stands ST1 is not healthy even if it does not grow as fast as the other.

Response 25: We have already described the reason in Response 7. All pines ages investigated are 32 ~ 36 a, so the different in stem density and growth may be caused by the site factors. The result in Table 3 shows that the C. kunyushanica prefer to plots with lower densities and higher tree height, DBH, and crown width, which consistent with the plant vigor hypothesis. We quite agreed with your description of vigor and health, and we know you're right. So, we have revised in L252 and L254.

Point 26: L250. I did not understand the sentence, especially 'after canopy closure'

Response 26: I’m sorry to make you confused. In text, we quoted this sentence to express that stand factors didn’t strongly affect the sawfly densities in some studies. In that reference, canopy closure means larch stands have closed canopies. We have revised this sentence as “In a previous study, Japanese larch (Larix leptolepis) density, tree height, DBH, and the proportion of larch stems were not strongly correlated with the prepupal densities of larch web-spinning sawflies (C. koebelei) when larch stands had closed canopies”.

Point 27: L276-279 and L293-294 you make big leaps from soils to insects. Soils affect first trees and tree quality affects insect performane. Thus, mention the role of trees, too.

Response 27: I am very sorry that what I have described is inappropriate, which makes it difficult for you to understand. The mature larvae of C. kunyushanica overwinter in soil from August to April of the next year. Many abiotic factors affect mortality of pupae in the soil, such as, teneral Dacus oleae suffered a higher mortality in hard soil. The overwintering larvae or pupae population of previous year is the basis of the adult populations this year. So, in L276-279, we want to describe the effect of soil porosity on pupae population.

As for sentences in L293-294, we have revised them as follows: So, similar to the nitrogen nutrient, soil organic matter enrich the nutrient resources and consequently increase the herbivore population. Besides, Nemer et al. showed that the mortality of C. tannourinensis prepupae is 100% in sandy soils, which lack organic matter. This also suggests that soil organic matter restricts the development and survival of the pupae of Pamphiliidae insects.

Point 28: L299 meaning of 'form of phloem N available' is not understandable, please explain

Response 28: Sorry, but we have revised the description as “It is possible that the deficient potassium nutrition of the host plant can enhanced nitrogen nutrition available to soybean aphids, which may increase the tendency of the aphid population to reach outbreak levels”.

Point 29: L299-301. Remove. No need to repeat the results and speculate with non-significant results.

Response 29: We have deleted the repeated results as you advised.

Point 30: L302-314. Shorten this chapter. You should not repeat the results here. You also discuss about the same points as earlier in the discussion, and you should combine them. You can shortly mention the benefit of RDA, but it is not necessary.

Response 30: The content in L302-314 are repeated, because we just discuss them earlier. We have deleted the repeated sentences as you advised.

Point 31: Conclusions. I think this is not a conclusion, but summary of results. Please, re-write.

Response 31: Thank you for your suggestion. We have rewritten the conclusion. First, we described the content and the main result of this research, then we put forward the key factors should be considered emphatically for controlling the C. kunyushanica populations, and finally we supposed the further research direction.  

Point 32: After your excellent introduction, I was waiting for discussion about the significance of the results from point of forest management and IPM from the modern perspective. I encourage you to discuss of these aspects, even shortly.

Response 32: Thank you for your suggestion about supplementing information on forest management. We have discussed the significant of our results by ecology control of forest pest in “Discussion” part.

Round 2

Reviewer 2 Report

I am happy with explanations and additions made by the authors. I still have some minor comments and suggestions for corrections that should be considered.

L20 replace correlation by relationship (you did not run correlation tests)

L 72. "needles when resources are rich"

For consideration

L70-71 sentence of Diprion pini is not necessarily needed. The next sentence L71-72 would fit with the next chapter where you describe biological performance of C. kunyushanica.

L71 C. kunyushanica mentioned first time, give the common name and whole scientific name there when first mentioned.

L195-203. If available, add references for the methods.

L266. You could simplify by starting "Other growth parameters of trees showed..."

Tables2-5. Consider accuracy of your measurements. Are your methods accurate enough to present results in an accuracy of two decimals or four significant digits? 

Tables3-5. I suggest a small change under the tables. Replace the last sentence as "The different letters indicate significant difference between the stand types"

L428-433. Your way of writing gives an impression that infestation was the reason for the growth changes. Change you style. "Stem density was lower and tree height,...., and stem volume higher in stands infested by C. kunyushanica." (Or did you also want to highlight that  the degree of infestation was affected in the starting chapter?  That may not be needed as you start with that on L434.) Formulate the sentences to be disussive, not a repetition of results with P-values.

443. "with higher tree height, DBH, crown width and so on" just repeats the previous sentence. Use more discussive style, "...denser and better growing JRP forests..." (?) On the other hand, the same topic is discussed later on the chapter. Consider compression.

455. De Somviele et al.

457-459. The sentence is not fluent. Consider this "This may because host plants contain reserves to restore their foliage, food does not become scarce, and thus, drive the sawflies out to less preferable plots."

475. Remove 'Table 5' - no need to refer figures and tables of results in the discussion

520-523. You should not explain results of soybean study, but discuss your own results. Reformulate the sentence and explain that similar process may have taken place in your study.

521. enhance (not enhanced)

532-540. I appreciate you added this discussion, although it is not totally fluent. Split the sentence starting "Meanwhile... year." "Thus, reducing the larvae or pupae is an approach preventing..." L540. Cleaning up forest litter sounds laborious. Do you mean after harvest? (Just an opinion, this could work for pest control, but removing litter is also removes natural fertlizers.)

547. emphatically is not a good word here, I think considered is enough

L654-656. You have mixed authors first and last names. It should be:

De Somviele, B.; Lyytikäinen-Saarenmaa, P.; Niemelä, P.

Author Response

Thank you again for your kind consideration. We have made some modification point-by-point according to your comments. If the article still needs some further modification, please let us known. We will do our best to revise it.

Point 1: L20 replace correlation by relationship (you did not run correlation tests)

Response 1: I’m sorry to make this mistake. We have replaced “correlation” by “relationship”.

Point 2: L 72. "needles when resources are rich"

For consideration

L70-71 sentence of Diprion pini is not necessarily needed. The next sentence L71-72 would fit with the next chapter where you describe biological performance of C. kunyushanica.

Response 2: “Needles when resource is rich” has been revised as “needles when resources are rich”. The sentence of Diprion pini has been removed. The next sentence on L71-72 has been adjusted to the next chapter, after “was identified as a new species in 1990”.

Point 3 L71 C. kunyushanica mentioned first time, give the common name and whole scientific name there when first mentioned.

Response 3: We have given the common name and whole scientific name there when C. kunyushanica first mentioned.

Point 4 L195-203. If available, add references for the methods.

Response 4: We have added references in the corresponding places.

Point 5 L266. You could simplify by starting "Other growth parameters of trees showed..."

Response 5: Thank you for your suggestion. We have revised that sentence as you advised.

Point 6 Tables2-5. Consider accuracy of your measurements. Are your methods accurate enough to present results in an accuracy of two decimals or four significant digits? 

Response 6: Thank you for your suggestion. As for “Tree height”, “Crown width”, and “DBH” in Table 3, we recorded one decimal in the field measurements, but they present two decimals just because we misrepresented them by the mean values of ST1-4 plots. However, We agree with your opinion on the accuracy of measurement, We have corrected that the values of the above three indices are one decimal. The data in Table 2 and Table 4 are calculated by formula, so it presents two decimals. The accuracy of the instrument for measuring zinc ions (Zn2+) is 0.001, but the content of Zn2+ in this study is lower, so it presents three decimals. For other soil chemical factors, we made them present two decimals refer to Gao et al. (2015).

Gao, R.H.; Shi, J.; Huang, R.F.; Wang, Z.; Luo, Y.Q. Effects of pine wilt disease invasion on soil properties and Masson pine forest communities in the Three Gorges reservoir region, China. Ecol. Evol. 2015, 5, 1702-1716.

Point 7 Tables3-5. I suggest a small change under the tables. Replace the last sentence as "The different letters indicate significant difference between the stand types"

Response 7: Thank you for your suggestion. We have changed the last sentence as you advised.

Point 8 L428-433. Your way of writing gives an impression that infestation was the reason for the growth changes. Change you style. "Stem density was lower and tree height,...., and stem volume higher in stands infested by C. kunyushanica." (Or did you also want to highlight that  the degree of infestation was affected in the starting chapter?  That may not be needed as you start with that on L434.) Formulate the sentences to be disussive, not a repetition of results with P-values.

Response 8: We have revised sentence as “Stem density was lower, while tree height, crown width, DBH, basal area at breast height, and stand volume were higher in stands infected by C. kunyushanica.”. I’m sorry to make you confused. Here, “the degree of infestation” means four stand types, but we have removed the sentence “The stand characteristics of JRP pure forests were different in stands with different infestation grades” on L434. The sentence on L430-431 has been revised as “Soil bulk densities and soil water content were higher in sawfly-non-infested stands than those in sawfly-infested types”.

Point 9 443. "with higher tree height, DBH, crown width and so on" just repeats the previous sentence. Use more discussive style, "...denser and better growing JRP forests..." (?) On the other hand, the same topic is discussed later on the chapter. Consider compression.

Response 9: Thank you for your suggestion. Considering the repetition of this sentence with the previous one and the later discussion on the chapter, we have compressed this sentence into “In this study, we observed that the PSI increased as the stem density decreased, and C. kunyushanica prefers to live in better growing (with higher tree height, DBH, crown width and so on) JRP forests.”.

Point 10 455. De Somviele et al.

Response 10: We have revised Bert et al as
De Somviele et al.

Point 11 457-459. The sentence is not fluent. Consider this "This may because host plants contain reserves to restore their foliage, food does not become scarce, and thus, drive the sawflies out to less preferable plots."

Response 11: There was something wrong with the writing of the sentence above , and we have revised it as you advised.

Point 12 475. Remove 'Table 5' - no need to refer figures and tables of results in the discussion

Response 12: We have deleted “Table 5” on L475.

Point 13 520-523. You should not explain results of soybean study, but discuss your own results. Reformulate the sentence and explain that similar process may have taken place in your study.

Response 13: Thank you for your suggestion. First, we have combined this paragraph with the previous one, because they were all discussed about the soil chemical properties. Then, we reformulated the sentence on L 520-523, and made it related to our own result.

Point 14 521. enhance (not enhanced)

Response 14: We have revised “enhanced” as “enhance”.

Point 15 532-540. I appreciate you added this discussion, although it is not totally fluent. Split the sentence starting "Meanwhile... year." "Thus, reducing the larvae or pupae is an approach preventing..." L540. Cleaning up forest litter sounds laborious. Do you mean after harvest? (Just an opinion, this could work for pest control, but removing litter is also removes natural fertlizers.)

Response 15: We have split the sentence into "Meanwhile... year." and "Thus, reducing the larvae or pupae is an approach preventing...". “Cleaning up” here means clean a little and properly, not thoroughly. However, your opinion is reasonable, and we revised it as “Thus, in these places, we can destroy soil chamber constructed by the mature larvae so as to prevent them from completing pupation.”.

Point 16 547. emphatically is not a good word here, I think considered is enough

Response 16: We have deleted “emphatically” in this sentence as you advised.

Point 17 L654-656. You have mixed authors first and last names. It should be:

De Somviele, B.; Lyytikäinen-Saarenmaa, P.; Niemelä, P.

Response 17: I’m sorry to mix authors first and last names, and we have revised it on L654-656.
